

# Assessing sampling and retrieval errors of GPROF precipitation estimates over The Netherlands

Linda Bogerd[1,2], Hidde Leijnse[2], Aart Overeem[2,3], and Remko Uijlenhoet[3]

[1]Hydrology and Quantitative Water Management, Wageningen University & Research, Wageningen, The Netherlands
[2]R&D Observations and Data Technology, Royal Netherlands Meteorological Institute (KNMI), De Bilt, The Netherlands
[3]Department of Water Management, Delft University of Technology, Delft, The Netherlands

**Correspondence:** Linda Bogerd (linda.bogerd@wur.nl)

**Abstract.** The Goddard Profiling algorithm (GPROF) converts radiometer observations aboard Global Precipitation Measurement (GPM) constellation satellites to precipitation estimates. Analyzing the accuracy of GPROF's estimates is vital to further improve the algorithm. Such analyses often use high-quality ground-based estimates as reference with a different spatial resolution. Often, the reference is resampled to match the satellite's resolution. However, the implemented sampling method to

5 simulate the satellite's resolution varies amongst studies, which limits the transferability of conclusions. Additionally, GPROF combines observations from various sensors and frequency channels, each with its own footprint size. Hence, uncertainties related to sampling are added on top of the uncertainty introduced when converting brightness temperatures to precipitation intensities. The contribution of sampling to the total amount of uncertainty remains unknown.

Here, we quantify the uncertainty related to sampling while analyzing the current performance of GPROF over the Nether-

10 lands during a four year period (2017–2020). In this area, shallow and light precipitation frequently occur. Both precipitation types are often subject to research, as both types are difficult to detect with space-borne sensors. Only GPROF estimates based on observations from the conical-scanning radiometers of the GPM constellation are used. We investigate the uncertainty related to sampling by simulating the reference precipitation as satellite footprints that vary in size, geometry, and applied weighting technique. The reference estimates are gauge-adjusted radar precipitation estimates from two ground-based weather

radars from the Royal Netherlands Meteorological Institute (KNMI). Echo top heights (ETH) retrieved from the same radars are used to classify the precipitation as shallow, medium, or deep.

The method used to spatially average the reference into a satellite footprint, i.e. using Gaussian weighting or the arithmetic mean, is found to exhibit a minimal influence on the retrieved estimate. The size of the sampled area is found to be the most influential. Still, the effect of using different footprint sizes cannot explain all the differences between the ground- and satellite-

20 based precipitation estimates. Additionally, the discrepancies between GPROF and the reference are largest for low ETH, while the relative bias between the different footprint sizes and implemented weighting methods increase with increasing ETH. Lastly, our results do not show a clear difference between coastal simulations and simulations over land. We conclude that the uncertainty introduced by merging different channels and sensors cannot fully explain the errors introduced by the retrieval algorithm. Hence, retrieval errors are found to be more prominent than sampling uncertainties, in particular for shallow and

25 light precipitation.



## 1    Introduction

Accurate and spatiotemporally uniformly distributed precipitation estimates are vital for both hydrological research and operational applications like weather forecasts and flood early water systems. Accurate estimates can be retrieved from well-established ground-based observations, such as weather radars and rain gauges. Their spatial coverage and representation,
however, is limited (Lorenz and Kunstmann, 2012; Saltikoff et al., 2019). This limitation can be overcome with the implementation of space-borne sensors. Yet, to date precipitation estimates retrieved from space-borne sensors are not as accurate as those derived from ground-based sensors (Chen and Li, 2016; Shen et al., 2020; Tang et al., 2020; Maggioni et al., 2022).

Determining the state-of-the-art accuracy of satellite-based estimates over different surface types and in various climates is crucial to further improve the performance of space-borne retrieval algorithms (Maggioni et al., 2016; Kim et al., 2017).
Retrieval algorithms use physical and/or statistical relations to convert space-borne observations to precipitation estimates at the Earth's surface. Their input is either indirectly (observation of cloud properties) or more directly (observation of hydrometeor properties) related to precipitation (Hou et al., 2008; Prigent, 2010; Kidd and Levizzani, 2011; Kidd and Huffman, 2011; Skofronick-Jackson et al., 2018).

The spatiotemporal resolution of indirect observations, retrieved from sensors aboard geostationary satellites, is higher than
that of more direct observations, retrieved from radiometers aboard low orbiting (LEO) satellites (Chang and Hong, 2012; Maggioni et al., 2016; Sun et al., 2018). Still, the latter are preferred for quantitative applications in meteorology and hydrology as precipitation retrieval from visible and infrared channels is based on cloud to precipitation relations. These statistical relations are location and time-dependent and generate precipitation estimates with poor accuracy (Lee et al., 2015; Kidd and Levizzani, 2019). In contrast, the upwelling radiation from the Earth's surface observed by radiometers is directly affected by precipitation
(Kidd and Huffman, 2011; Maggioni et al., 2016; Kidd et al., 2021b). Precipitation increases microwave emissions measured by the lower frequency channels and decreases microwave emissions measured by the higher frequency channels (Kummerow, 2020).

One algorithm that converts microwave emissions, often expressed as brightness temperatures (Tb), to precipitation estimates is the Goddard Profiling precipitation retrieval algorithm (GPROF) (Kummerow et al., 2001, 2015). GPROF is a Bayesian
algorithm that uses an a-priori database of Tbs, hydrometeor profiles, and surface precipitation estimates. GPROF's database is built on observations from the two sensors aboard the Global Precipitation Measurement (GPM) core-satellite: the GPM Microwave Imager (GMI) and the dual-precipitation radar (DPR) (Hou et al., 2014; Skofronick-Jackson et al., 2018). GMI is a radiometer equipped with thirteen frequency channels. The combination of Tbs measured by each channel is matched to simultaneous DPR hydrometeor profiles and surface precipitation estimates (Randel et al., 2020).

Although calibrated on GMI, GPROF is able to convert observations from all radiometers aboard GPM constellation satellites into precipitation intensities (Kummerow et al., 2015; Randel et al., 2020). However, the size of an area scanned by a radiometer, also referred to as footprint, varies with sensor and frequency channel (Guilloteau et al., 2017). For instance, the diameter of the footprint associated with the 19 GHz channel, which is often used as 'reference' resolution (You et al., 2020), is more than twice as large in both across- and along-scan direction compared to the footprint associated with the 89 GHz chan-



nel. Hence, merging various channels and sensors inevitably implies merging observations with different spatial resolutions. This difference in resolution introduces uncertainty as precipitation is highly variable in space and time (Foufoula-Georgiou et al., 2014; Cristiano et al., 2017; Leth et al., 2021).

Up to now, research has mostly focused on the uncertainty related to assumptions in the retrieval algorithm to improve detection and accuracy. An example of a persistent challenge is the retrieval of shallow and light precipitation (Liu and Zipser,

2014; Ferraro et al., 2013; Kidd et al., 2021a; Hayden and Liu, 2021). As mentioned, large water or ice particles interact with the upwelling radiation. This interaction is weaker for shallow and light precipitation (Casella et al., 2015; Kummerow et al., 2015). Analyzing brightness temperatures of the individual frequency channels during shallow and light precipitation events reveal what radiometers do observe when those types of precipitation occur. However, as explained before, each channel is associated with a different spatial scale due to the associated footprint size. Hence, before analyzing the brightness temperatures related

to each channel, the uncertainty related to sampling needs to be identified.

First, we briefly evaluate the most recent version of GPROF, V07, as its performance over mid- to high latitudes has, to the best of our knowledge, not been evaluated yet. Second, we analyze to what extent the evaluation of space-borne estimates is affected by the sampling pattern used to align the reference and space-borne observations. Additionally, this study evaluates the uncertainty introduced when merging various footprint sizes associated with the different channels and radiometers using

only reference estimates. Lastly, we determine how different characteristics, such as the vertical extent of precipitation or the proximity of the coast, affect the uncertainty related to sampling. This uncertainty is analyzed by simulating the footprints of the three conical scanners that belong to the GPM constellation. The footprints are simulated using 1 km × 1 km gauge-adjusted radar precipitation estimates provided by the Royal Netherlands Meteorological Institute (KNMI). The Netherlands, a coastal country where shallow and low-intensity precipitation frequently occurs, is used as study area from January 2017 to

December 2020. Ground-based echo top heights (ETH) are used to classify the vertical extent of precipitation within a certain footprint.

## 2  Measurement and methods

### 2.1  Data

The three precipitation datasets used in this study were all available over the research area, the Netherlands (50.78°–53.68°N,

3.38°–7.38°E; 35 000 km$^2$), during the entire studied period, from 1 January 2017 to 31 December 2020. Each dataset is briefly described in the following subsections. An elaborate description of precipitation occurring in the research area with similar reference data partly overlapping the current research period can be found in Bogerd et al. (2021).

### 2.1.1  Satellite observations: GPM constellation conical scanning radiometers

The core satellite of the Global Precipitation Measurement mission (GPM) was launched in 2014. GPM aims to increase

both the availability of precipitation data over ungauged areas as well as the understanding of precipitation processes (Hou





et al., 2014; Skofronick-Jackson et al., 2017; Skofronick-Jackson et al., 2018). To achieve these aims, the mission consists of a constellation of satellites carrying radiometers and a "core-satellite". The core satellite carries both a radiometer with a broad spectrum of frequencies (GPM microwave imager: GMI) and a precipitation radar (DPR). This setup provides the opportunity to couple simultaneous radiometer observations and vertical precipitation structures from space. These simultaneous observations are used as input for the GPROF algorithm (Kummerow et al., 2015).

GPROF converts brightness temperatures retrieved from radiometers onboard GPM constellation satellites into precipitation estimates. GPROF is parametric: it works with all these radiometers as long as the characteristics and channel errors of each sensor are known. The algorithm is based on a Bayesian approach and uses an a-priori database of observed cloud and hydrometeor profiles, based on DPR observations. These profiles are matched with simulated radiances. The radiometer observations are compared to simulated radiances to gain a weighted sum from which precipitation estimates are computed. More details about GPROF can be found in Kummerow et al. (2015); Passive Microwave Algorithm Team Facility (2022); Randel et al. (2020). This study focused on the three conical scanning radiometers contributing to GPM: SSMIS, AMSR-2, and GMI. Their footprint sizes are shown in Tab. 1. The conical scanners are the most important radiometers within IMERG, GPM's gridded precipitation product based on GPROF estimates. IMERG selects conical scanners in case of simultaneous radiometer overpasses over a certain area.

### 2.1.2 Ground-based precipitation estimates: gauge-adjusted radar

The Royal Netherlands Meteorological Institute (KNMI) offers a high-quality gridded precipitation product at a spatial resolution of $\sim 1\ km^2$ and a 5-min temporal resolution. This product is based on composites of two polarimetric C-band radars. For this product, precipitation is retrieved every 5 minutes by using data from scans at 3 ($0.3°$, $1.1°$, and $2.0°$) out of 16 elevations from both radars. After the two radar composites are combined, the two rain gauge networks from KNMI, involving 31 automatic and 325 manual gauges, are used to adjust the radar precipitation estimates. Elaborate descriptions about this dataset can be found in Overeem et al. (2009a, b, 2011).

### 2.1.3 Ground-based echo top height observations: radar

Ground-based radar echo top height (ETH) data was used to classify precipitation based on its vertical extent. This classification allows to study both to what extent the precipitation height influences the performance of GPROF and how ETH is related to precipitation variability within a certain footprint size. The ETH is defined as the maximum height at which a particular reflectivity threshold, in this case 7 dBZ, is exceeded.

The ETH observations were retrieved from the same ground-based C-band radars described in the previous subsection. However, the ETH product is based on all fifteen elevations (ranging from $0.3°-12.0°$). The authors are aware of the deficiencies associated with this product. For instance, the low detection threshold of 7 dBZ combined with residual clutter and overshooting that occurs at large distances from the radar can induce unrealistically high or low ETH values. Hence, ETH observations below 1 km and above 15 km were removed before further analysis. A footprint was classified as low when $1\ km \leq \text{ETH} < 3\ km$, medium when $3\ km \leq \text{ETH} < 6\ km$, and high when $\text{ETH} \geq 6\ km$. The allocation of a footprint to a certain class is based on



the averaged ETH value within a footprint of the size associated with the 19 GHz channel. More information about the ETH
product and its evaluation can be found in Beekhuis and Holleman (2008); Aberson (2011).

## 2.2  Spatiotemporal matching

During the study period, all overpasses by the three conical scanners with more than eight pixels over the land surface of the
Netherlands were selected. The coordinates provided along with the satellite observations represent the center of the pixel.
Subsequently, using the scan position of a particular pixel, the orientation of the elliptic shaped footprint was determined. The
scan pattern of the GMI scanner is shown in Hou et al. (2014) (Fig. 2, an example of one GMI scan is shown in light blue).
The size of each footprint depends on the satellite and frequency. This study used the dimensions associated with the 19 and
89 GHz channels. These channels are considered crucial for precipitation observations as hydrometeors interact with radiation
at these frequencies (Stephens and Kummerow, 2007; Kummerow, 2020) This assumption is also used within GPROF (Passive
Microwave Algorithm Team Facility, 2022). Because GPROF assumes the footprint sizes of GMI, GMI's dimensions are used
in this study. The high-resolution ground-based observations within each simulated footprint were averaged using either the
arithmetic mean or Gaussian weighting.

Additionally, pixels with center coordinates within 40 km distance of the coast were identified. The coast is highlighted
because the accuracy of space-borne precipitation retrieval over coastal areas is often reduced compared to its accuracy over
land or sea/ocean (Kubota et al., 2009; Mega and Shige, 2016; Munchak and Skofronick-Jackson, 2013). This reduction
is attributed to the sudden change in background radiation, which is different for land and sea/ocean. Hence, the background
radiation can vary within a footprint close to the coast. However, precipitation dynamics and the difficulties to correctly capture
this could also be the reason for a change in accuracy, as the temperature difference between the coast and land affects the
occurrence of precipitation (and the associated precipitation types). By studying the reference estimates sampled on different
scales while taking into account the proximity of the coast, we can study the sensitivity of precipitation events and their
intensity as a function of coastal distance and either reject or confirm this relationship.

## 2.3  Validation

Established metrics were used to assess the performance of GPROF. The relative bias (RB) was calculated to determine the sign
and magnitude of the bias between the evaluated product (GPROF) and the reference (ground-based precipitation estimates).
A positive (negative) value indicates that the evaluated product overestimates (underestimates) the precipitation intensity com-
pared to the reference. The normalized mean absolute error (NMAE) was calculated to demonstrate the overall error magnitude,
normalized by the average of the reference values. If NMAE equals 1, GPROF values are, on average, off by the same mag-
nitude as the reference mean. Due to its normalization, NMAE allows to compare the performance amongst different ETH
classes (higher ETH is often associated with higher precipitation intensities).



The RB and NMAE are defined as follows:

$$\text{RB} = \frac{\sum\limits_{i=1}^{n}(R_{\text{evaluated},i} - R_{\text{reference},i})}{\sum\limits_{i=1}^{n} R_{\text{reference},i}}, \tag{1}$$

$$\text{NMAE} = \frac{\sum\limits_{i=1}^{n}|R_{\text{evaluated},i} - R_{\text{reference},i}|}{\sum\limits_{i=1}^{n} R_{\text{reference},i}}, \tag{2}$$

where $n$ represents the number of pixels available, in both space and time. Additionally, the probability of detection (POD) was used to measure GPROF's ability to distinguish wet and dry pixels, with a threshold of 0.1 mm/hr. It is important to note that the probability of false alarms (POFA) could not be calculated. Only estimates corresponding with valid ETH observations were selected. A valid ETH value automatically implies that the ground-based radar measured rain. The POD is defined as:

$$\text{POD} = \frac{\text{hits}}{\text{hits} + \text{misses}}, \tag{3}$$

where "hit" means that both GPROF and the reference identify a pixel as "rainy" (exceeding 0.1 mm/hr) and "miss" means that the reference identifies a pixel as rainy (exceeding 0.1 mm/hr) while GPROF identifies the pixel as dry (intensity lower than 0.1 mm/hr).

## 3 Results

### 3.1 Evaluation of GPROF's performance

First, the performance of GPROF is determined. Coupled GPROF and reference estimates, as a function of their proximity to the coast, are shown in Fig. 1. Coupled estimates deviate from the 1:1 line in both directions, meaning GPROF both under- and overestimates precipitation intensity compared to the reference. In general, however, GPROF underestimates precipitation intensity as the reference mean is higher. This result is independent of the observation sensor (SSMIS, AMSR-2, or GMI) or the proximity of the coast. SSMIS-based GPROF estimates close to the coastal area have the smallest discrepancy: 0.89 mm/hr vs. 1.13 mm/hr according to the reference. GPROF can only explain 34%–37% of the variance observed in the reference estimates, which is even reduced to only 23–27% for estimates retrieved from GMI and AMSR-2. For SSMIS and GMI, the underestimation of GPROF is worse over land, while for AMSR-2 the lowest performance is over the coastal area, especially for low intensity events (Fig. 1, upper panel, left).

*Figure 1 approximately here.*

Figure 2 explores the results of Fig. 1 in more detail by evaluating GPROF's performance as a function of vertical extent. The metrics are calculated with the reference sampled to the footprint size associated with the 19 GHz channel of GMI (circles) or the observing sensor (stars). Both are considered to evaluate the sensitivity of GPROF's performance concerning



the implemented method used to sample the reference. However, only observations exceeding the 0.1 mm/hr threshold are considered. Due to different footprint dimensions, the number of observations that exceed this threshold might differ. Hence, for SSMIS and AMSR-2 two GPROF means are shown: GPROF's mean based on the observations that are coupled to reference estimates sampled to the footprint size associated with the GMI 19 GHz channel (triangles) or to the 19 GHz channel of the observing sensor (plusses).

Both the reference and GPROF mean increase with increasing ETH. Independent of the observation sensor, corresponding ETH, or the size of the sampled area, GPROF's mean is low compared to the reference mean and the RB is negative. An exception are GMI observations associated with high ETH. For these observations, the reference and GPROF mean values are similar and RB is close to zero. Additionally, GMI observations hardly ever miss precipitation associated with a high ETH, illustrated by the POD being close to 1. In contrast, the POD for shallow precipitation does not exceed 0.6 for any of the sensors, indicating that GPROF's ability to correctly detect wet pixels for shallow precipitation is not higher than 60%. GPROF's enhanced performance for estimating precipitation intensity associated with high ETH compared to those associated with low ETH is less evident from the $R^2$ and NMAE. These two statistics are less dependent on the precipitation intensity, which increases with increasing ETH. The intensity is more reflected in both RB and the mean values.

*Figure 2 approximately here.*

Figure 2 shows, as expected, that the influence of sampling is especially relevant for SSMIS observations. The SSMIS footprint is much larger, whereas ASMR-2 and GMI have similar footprint sizes (Tab. 1). The bias (both NMAE and RB) increases while $R^2$ decreases when evaluating SSMIS observations against the reference sampled at GMI resolution instead of its native footprint size. This result is confirmed by Fig. 3, which shows the cumulative distribution functions of the occurrence of precipitation intensities (CDF, solid lines). Figure 3 aims to highlight the effect of using different sampling methods on the 'estimated' precipitation intensity. The reference sampled to AMSR-2 resolution and GMI resolution are plotted on top of each other, implying similar results. The occurrence and contribution of high intensity precipitation are reduced for reference values sampled to SSMIS (lightblue) resolution (lower panels). In general, maximum intensities increases with increasing ETH both for GPROF and the reference.

*Figure 3 approximately here.*

### 3.2 Sampling sensitivity analysis

The first three figures focused on GPROF's performance and provided some first results concerning the influence of sampling. The remainder of this study concentrates on sampling only. Hence, all results are based on ground-based reference estimates averaged on various footprint sizes and geometries. Figure 4 is based on GMI coordinates and resolution and compares averaged estimates using four different sampling methods. The y-axis represents, from left to right, averages of the high-resolution estimates calculated using a 19 GHz gaussian circle, 19 GHz spatial ellipse (arithmetic average), and 89 GHz gaussian ellipse. With 'gaussian' we refer to gaussian weighting.

The averaged estimates calculated using uniform weights are on the 1:1 line (middle panel) and $R^2$ is 1 (mid panel). Although a circle results in more noise (left panel) compared to the choice of weighting, the scatter is still limited compared to the scatter



observed in Fig. 1. The size of the ellipse results in the largest difference amongst the references (right panel). Estimates based on the 89 GHz channel are skewed towards higher values compared to those based on the 19 GHz channel footprints. This finding is in agreement with the results of the bottom panels of Fig. 3. The lower panel features observations associated with a shallow vertical extent. The deviations seem smaller than the deviations shown in the upper rows, likely due to the lower precipitation intensities related to shallow events. In contrast, the $R^2$ is lower for shallow precipitation. The number of misses

varies amongst the sampling methods, indicating the sampling method could (slightly) affect the POD. Again, the largest effect is the size of the sampling area, thus the channel footprint size (right panel). Still, the POD is higher than the POD of GPROF found in Fig. 2. GMI and AMSR-2 have similar footprint dimensions, resulting in similar results (not shown).

*Figure 4 approximately here.*

Figure 5 is similar to Fig. 4 but considers footprint dimensions based on SSMIS channels. Spatial-weighted ellipse vs.

225 gaussian-weighted ellipse is not shown as the results were similar to the middle panel of Fig. 4, indicating limited deviations between the two sampling methods. Hence, more emphasis has been put on the dimensions of the sampled area.

The differences between the SSMIS 19 GHz and SSMIS 89 GHz channels are larger than the differences between the SS-MIS 19 GHz and GMI 19 GHz channels. Additionally, more observations are missed and $R^2$ is lower ($R^2$=0.53 vs. $R^2$=0.61). Although both footprint dimensions are smaller than the footprint dimensions associated with SSMIS 19 GHz channel, GMI's

length-width proportions are more similar to the proportions of the 19 GHz SSMIS channel. Considering only shallow observations yields similar conclusions (lower panels). The relative amount of misses is high (up to 43%) compared to the upper row, comparable to GPROF's POD for shallow precipitation that was found to be independent of footprint size in Fig. 2.

*Figure 5 approximately here.*

Fig. 5 shows large deviations in sampled reference precipitation estimates using the footprint sizes associated with the

235 19 GHz and 89 GHz channels. Both channels are considered important for precipitation retrieval, especially for the GPROF algorithm. To obtain more insight into the circumstances for which the uncertainty related to merging the two channels would be largest, the observations are evaluated in more detail in Fig. 6. The observations are studied as a function of vertical extent, distance from the coast, and season. The vertical extent has a clear seasonal cycle (right panels): the occurrence of precipitation with a shallow vertical extent is highest in winter, both relative as well as in absolute numbers. High ETH frequently occurs in

the summer season while its occurrence is limited during the other seasons, especially in winter. Both the standard deviation (std) and RB increase with increasing height, independent of season and surface type. The $R^2$ does not exceed 0.65, except for high ETH during winter. The RB is always positive, meaning simulations with a smaller footprint result in higher precipitation estimates.

*Figure 6 approximately here.*

Figure 7 is similar to Fig. 6, but based on the overpasses of the three sensors. All observations are sampled to footprint sizes associated with GMI's 19 GHz or 89 GHz channels to increase the sample size (the results are similar when only the overpasses used in Fig. 6 are used, not shown). In agreement with Fig. 4, Fig. 7 indicates larger $R^2$ values compared to Fig. 6 due to the smaller difference in footprint size. Additionally, RB values are 10 times smaller compared to Fig. 6 and $R^2$ is never below



0.78. The footprint associated with GMI is small compared to the SSMIS footprint, resulting in a larger std compared to Fig. 6, especially for the high ETH regime.

*Figure 7 approximately here.*

## 4 Discussion

First, this study analyzed the performance of GPROF V07, the most recent version GPROF. This version seems to perform better than its predecessor, GPROF V05, both in terms of detection as well as the accuracy of the intensity. V05 was known to either miss shallow events (Kidd et al., 2018; You et al., 2020; Tan et al., 2022) or highly overestimate the intensity over mid- to high latitudes (O et al., 2017; Bogerd et al., 2021). The large overestimations found in V05 seems reduced in version V07 and the POD seems improved as well (light blue geometries in Fig. 2). Yet, the POD associated with shallow events remains low in V07. Furthermore, V07 is still challenged by light precipitation, in line with the results of Pfreundschuh et al. (2022). In general, however, Pfreundschuh et al. (2022) found a higher performance of V07 compared to the results presented in this study. This difference can (partly) be attributed to the implemented reference data. Pfreundschuh et al. (2022) used the GPM combined algorithm, which is based on DPR and GMI observations, as reference, while both DPR and GMI contribute to GPROF. Furthermore, their study area does consider various climates, as their study has a global focus. The difficulties associated with shallow precipitation are most likely related to the weak signal associated with stratiform shallow events (Tan et al., 2022), a common precipitation type in the Netherlands.

Coastal areas are challenging for space-borne radiometer precipitation retrieval due to the sudden change in background radiation (McCollum and Ferraro, 2005; Mega and Shige, 2016; Petty and Bennartz, 2017). Hence, we tested the sensitivity of the results when taking into account the proximity of the coast. Footprints were classified as 'coastal area' when its coordinates are within a 20 km or 40 km radius from the coastline. Independent of the implemented distance, the performance of GPROF is not significantly worse over the coastal region (Fig. 1). These results suggest that the additional coastal categories based on the percentage of water (Passive Microwave Algorithm Team Facility, 2022) improved GPROF's performance over coastal areas.

Furthermore, the major conclusions about GPROF's performance are consistent amongst the three evaluated sensor types within the GPM constellation. AMSR-2 has the lowest POD and the highest error metrics for shallow precipitation (Fig. 2). The score of AMSR-2 associated with these precipitation types decreases even further when excluding pixels within 40 km of the coast (not shown). The footprint size is eliminated as a possible cause since AMSR-2 and GMI have comparable footprint sizes, while SSMIS's footprint is much larger. Instead, the poor performance of AMSR-2 is most likely related to the limited number of high-frequency channels, as the highest frequency channel of AMSR-2 is 89 GHz. Especially the higher-frequency channels are considered important over land where ice-scattering properties are used to calculate precipitation from Tb observations (Shin and Kummerow, 2003; You et al., 2017; Wang et al., 2018) within the GPROF algorithm (Kummerow et al., 2015).

Figures 4, 5, 6, and 7 focus on the effect of sampling. The difference in implementing either gaussian weighting or uniform weighting is found to be negligible, especially for shallow observations. Hence, publications assuming circles or spatially averaged ellipses should yield comparable conclusions. Area is found to be the most important. Hence, it is expected that





merging the different frequencies and sensors results in uncertainty added to the precipitation estimates of GPROF, as all observations are converted to GMI's 19 GHz footprint dimensions. Each sensor and frequency channel is associated with its own footprint, while GPROF assumes the footprint size associated with the 19 GHz channel. From Fig. 6 it becomes apparent
that both the mean and standard deviation increase with ETH, while they are only to a lesser extent affected by the implemented sampling methods. When we evaluate the effect on the performance of real instead of simulated GPROF precipitation estimates (Figs. 2 and 3), it becomes apparent that GPROF's performance is mostly influenced by ETH and thus the effect of sampling can only partly explain the discrepancy between GPROF and the reference observations. Additionally, this discrepancy is largest for low ETH when the effect of footprint size is found to be minimal. Hence, improving the accuracy of shallow and light intensity
precipitation estimates from space-borne observations should be addressed by improving the (physical relations within) the algorithm. For instance, the DPR is used to match the radiometer observations to radar observations. This dependency can result in inaccuracies if the DPR is not able to capture shallow precipitation.

A brief evaluation of the DPR's performance is shown in Fig. 8. This figure clearly shows the DPR has difficulties in both detecting and accurately quantifying the amount of shallow precipitation. However, as mentioned before, DPR observations
are used to calibrate GPROF. Hence, future studies are recommended to focus more on the physical characteristics of shallow precipitation and how to improve their estimates using space-borne sensors.

*Figure 8 approximately here.*

## 5   Conclusion

Radiometers are essential to provide a global precipitation products based on uniformly distributed measurements from low
earth orbit spatial platforms. Hence, a lot of effort is put into addressing persistent challenges and reducing uncertainties associated with algorithms that convert brightness temperatures into precipitation estimates. Yet, these algorithms will always be associated with a certain amount of uncertainty due to the merging of various channels with different footprint sizes for shallow and light precipitation over The Netherlands (53°N). This study provides insight into the magnitude of this uncertainty through sampling high-resolution estimates using different geometries and footprint sizes.
GPROF, the retrieval algorithm of the Global Precipitation Measurement mission (GPM) that converts brightness temperatures into precipitation estimates, was first evaluated to be able to quantify the discrepancy between space-based radiometer estimates and ground-based radar estimates. The $R^2$ between GPROF and the reference varies between 0.23 and 0.37. Additionally, GPROF has difficulties to detect precipitation with a shallow vertical extent. As a next step, simulated footprints based on reference data used to evaluate GPROF were analyzed. This analysis provided insight into the uncertainties related to the
combination of various channels, sensors, weighting methods and their corresponding footprint dimensions.

The implemented weighting method and chosen geometry (circle or ellipse) was found to have a limited effect on the simulated footprints. Although the size of the footprints has a larger effect on the values of the retrieved estimates, it can not fully explain the discrepancies between GPROF and the reference estimates. Additionally, GPROF's relative bias is large for shallow ETH while the relative bias between simulated estimates based on different sampling areas increases with ETH. We



conclude that most of the uncertainty is related to the retrieval algorithm. At the same time, this study raises awareness about the inevitable uncertainties introduced when merging various channels and sensors. Hence, our results are also relevant for choosing appropriate footprint sizes when comparing to reference data.

*Code and data availability.* All data from GPM can be accessed at https://gpm.nasa.gov/data. All data from KNMI can be accessed at https://dataplatform.knmi.nl/

*Author contributions.* **Linda Bogerd**: conceptualization, methodology, data, formal analysis, visualization and writing original draft. **Hidde Leijnse**: conceptualization, methodology, supervision, and review. **Aart Overeem**: conceptualization, methodology, supervision, and review. **Remko Uijlenhoet**: conceptualization, methodology, supervision, and review.

*Competing interests.* The authors declare that they have no known competing financial interests or personal relationships that could have appeared to influence the work reported in this paper.

*Acknowledgements.* We acknowledge financial support from the Dutch Research Council (NWO) through project ALWGO.2018.048. Furthermore, we would like to thank Claudia Brauer for her broader view on this manuscript and inspiring discussion on the figures. Lastly, we would like to acknowledge Lisa Milani for sharing her expertise on shallow precipitation, in particular snowfall.



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



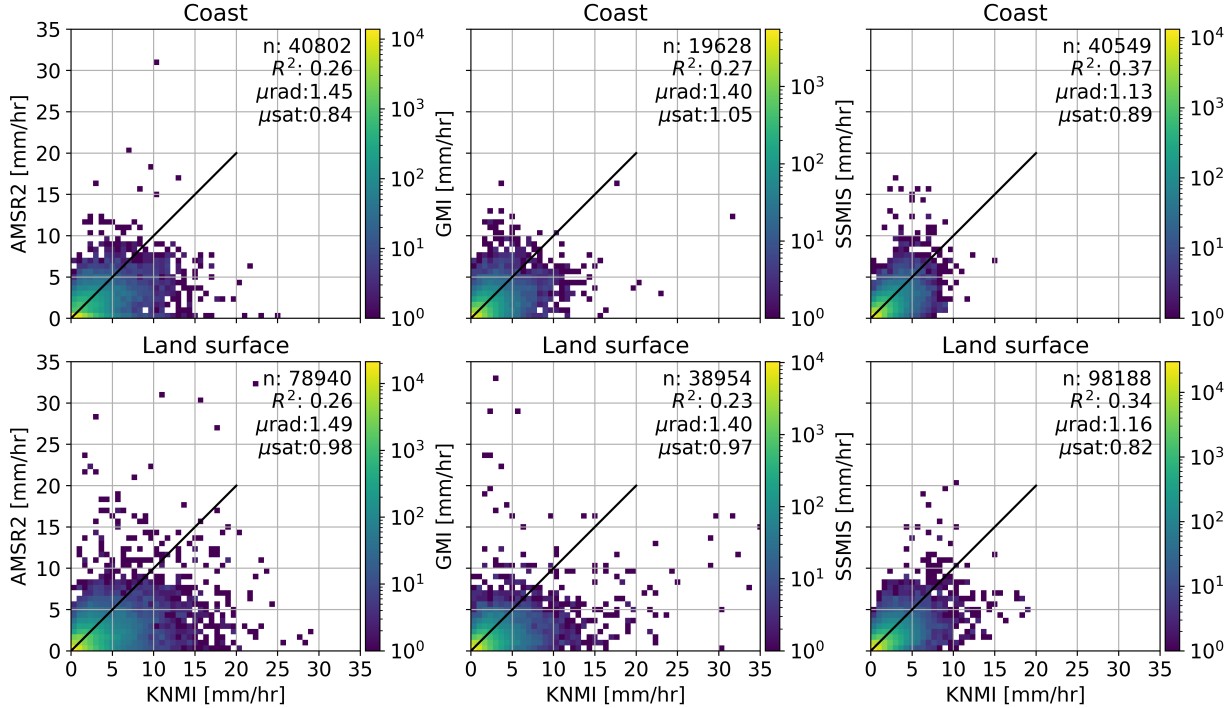

**Figure 1.** Scatter density plots of GPROF vs reference estimates for the entire study period (January 2017–December 2020). The first row shows observations within 20 km distance from the coast, the second the remaining observations over land. The reference estimates are sampled to the footprint corresponding to the 19 GHz channel associated with the sensor (i.e. AMSR-2, GMI, SSMIS). Only paired observations where both references exceed 0.1 mm/hr (hits) and $1\ km \leq \text{ETH} < 15\ km$ are considered.

**Table 1.** Resolution of the three conical scanning radiometers implemented in this study.

| Sensor | Along scan [km] | Cross scan [km] |
|---|---|---|
| GMI 19 GHz | 10.9 | 18.0 |
| GMI 89 GHz | 4.4 | 7.2 |
| AMSR2 19 GHz | 14.0 | 22.0 |
| AMSR2 89 GHz | 3.0 | 5.0 |
| SSMIS 19 GHz | 45.0 | 74.0 |
| SSMIS 89 GHz | 13.0 | 16.0 |



**Figure 2.** Statistics of GPROF for the three sensors using reference estimates averaged on the footprint associated with the GMI 19 GHz channel (circles) or each sensor's own 19 GHz channel (stars). The triangles (GMI 19 GHz channel) and pluses (own sensors 19 GHz channel) in the left upper panel represent GPROF's mean. Additionally, the vertical extent of precipitation was taken into account (different blue shades). The statistics are based on all overpasses during the study period. Except for the contingency metric (POD), paired observations where both references exceed 0.1 mm/hr (hits) and $1 \, km \leq \text{ETH} < 15 \, km$ are considered.



**Figure 3.** Cumulative distribution functions of precipitation intensity occurrence (CDF: solid line) and volume (CDFv; dashed line) for GPROF (black) and the reference (light blue) using the reference estimates averaged on the footprint associated with the GMI 19 GHz channel. Additionally, similar to Fig. 2, the results are also shown using the footprint associated with the 19 GHz channel of SSMIS and AMSR-2 (GPROF: grey, reference: darkblue). The sampling method and time period are the same as Fig. 2). Only paired observations where both references exceed 0.1 mm/hr (hits) and $1\ km \leq$ ETH $< 15\ km$ are considered. The CDFs are calculated with a logarithmic bin width. The rows represent the different sensors, the columns the different ETH classes.



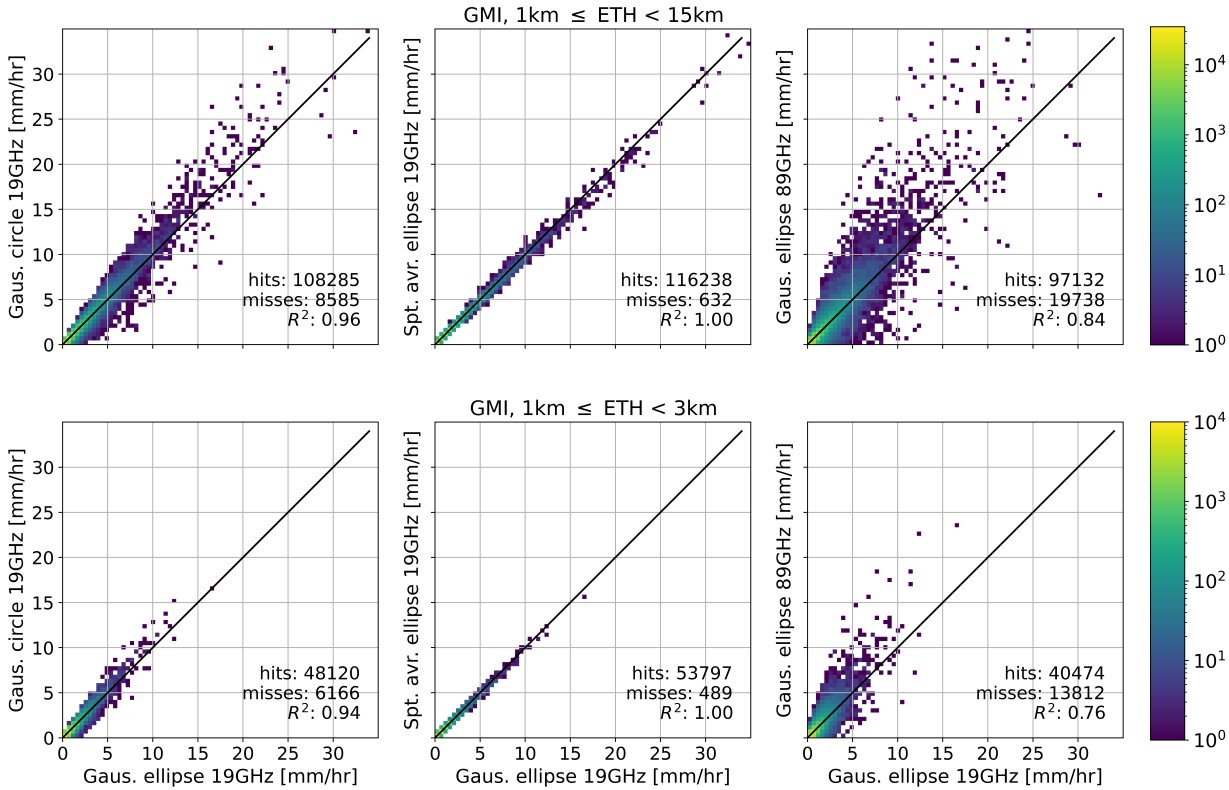

**Figure 4.** Scatter density plots of simulated GMI observations using four sampling methods for the entire study period (January 2017–December 2020). Only paired observations where both references exceed 0.1 mm/hr (hits) and $1\,km \leq \mathrm{ETH} < 15\,km$ (upper panel) or $1\,km \leq \mathrm{ETH} < 3\,km$ (bottom panel) are considered. All scatter density plots consider the same observations. The number of hits varies due to misses by the sampling method on the y-axis. The number of misses is shown in the bottom right of each subfigure, together with the number of hits and $R^2$.



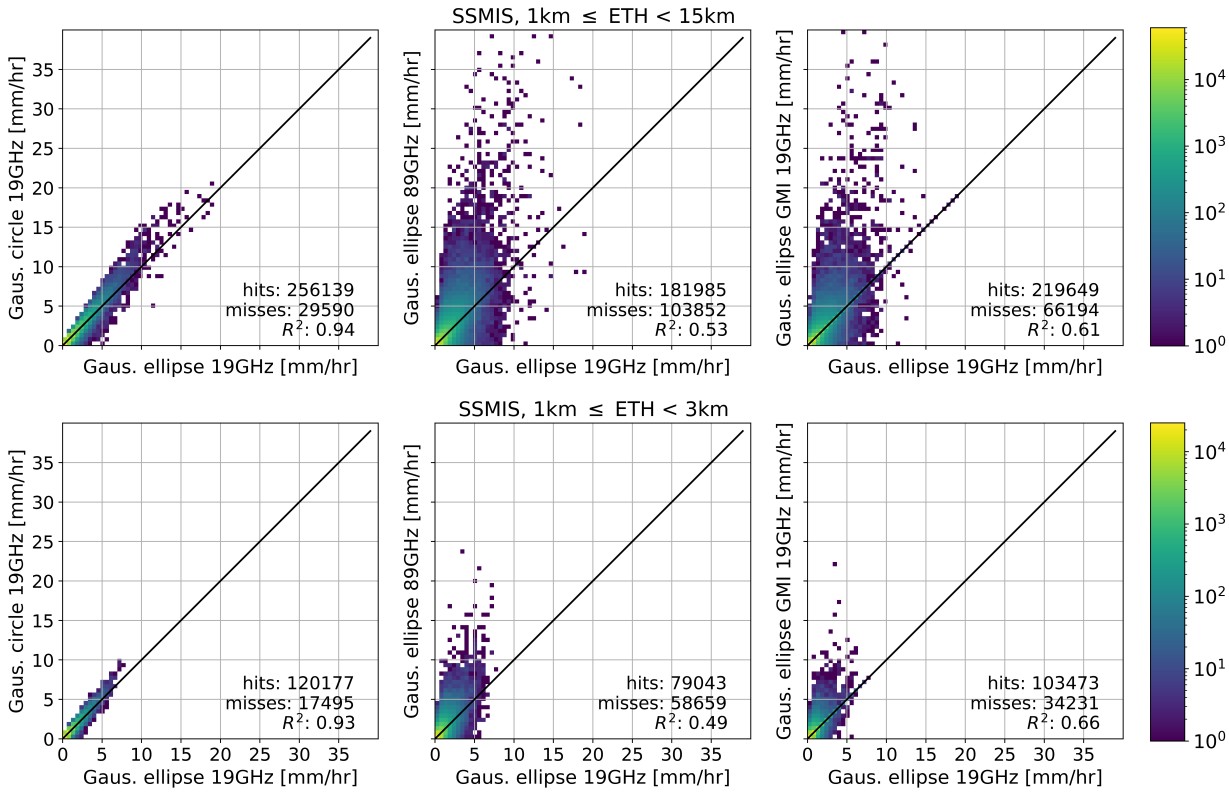

**Figure 5.** Similar to Fig. 3 using SSMIS observations and corresponding footprint sizes.



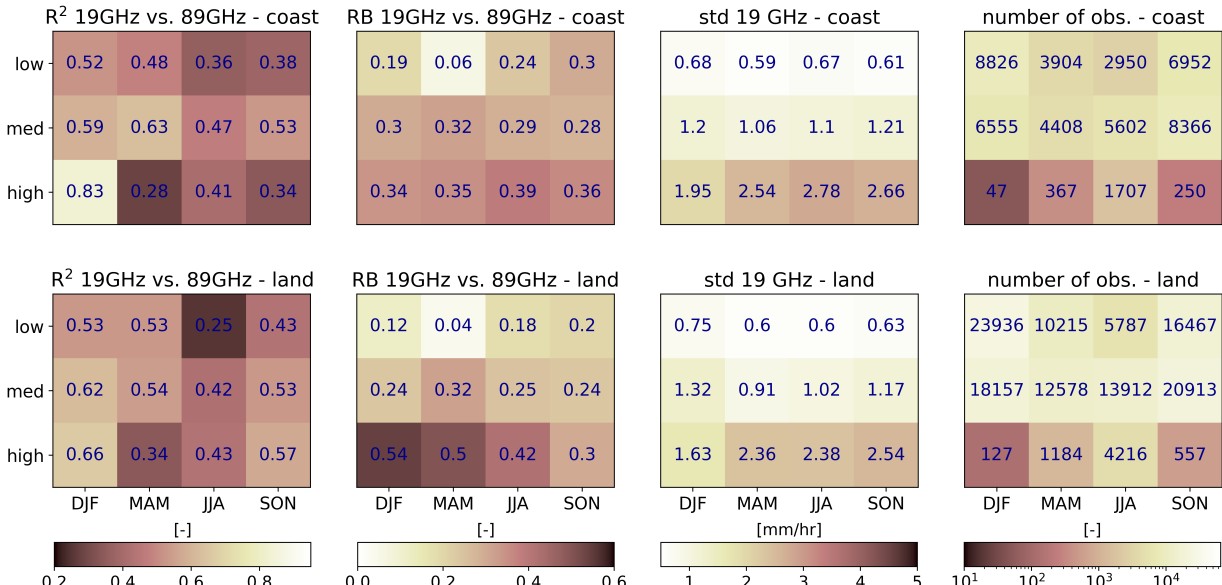

**Figure 6.** $R^2$, relative bias (RB), standard deviation (std), and number of observations (n) for SSMIS observations and corresponding footprint sizes. Same observations as used in Fig. 5. The upper panels represent the statistics of all observations within 20 km distance of the coast, the lower panels those of the other observations (i.e. over land). The statistics are shown as a function of ETH and season. The background color represents the value of that particular cell. Additionally, the value itself is presented in each cell in darkblue.

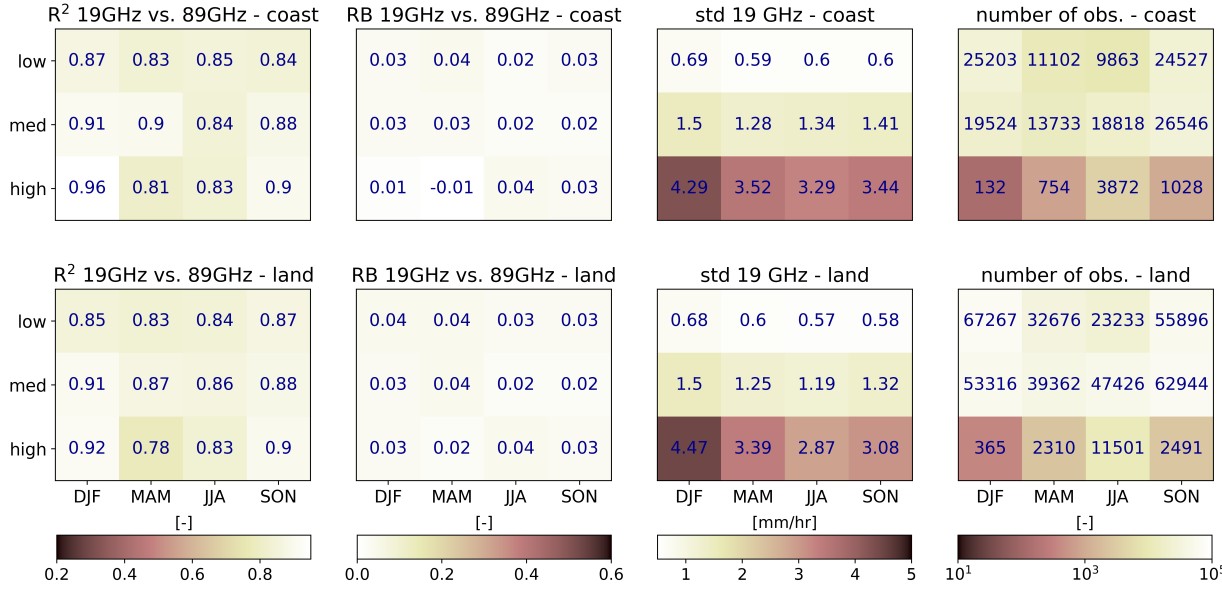

**Figure 7.** Similar to Fig. 6, but now including all sensors and using the footprint dimensions of GMI.





**Figure 8.** Same as Fig. 2, but for the DPR. FS means "Full-Scan" (both Ku- and Ka-band over the entire Ku-band swath), HS "High-frequency Scan" (observations from the Ka-band over the Ka-band swath, which is smaller than the Ku-band swath.