# Peer review of "Assessing sampling and retrieval errors of GPROF precipitation estimates over The Netherlands"

_EGUsphere, 2023_

## Referee Comment (RC2)

This interesting study is well developed and executed. The topic of estimating/comparing precipitation/rainfall by different remote sensing-based data sources is quite challenging. My comments mostly are textual where further detailing and description of used terms is advised to increase readability. As an example, *sampling* is widely used but it is not always clear how to read the meaning of the term. E.g. sampling method, sampling pattern, sampling sensitivity, sampling area are used but remain somewhat undefined. Also the term pixels is used but also undefined in terms of size shape. It would be good to add a figure that shows how spatial units of sampling overlays the pixel structure of the SRE products. Also *uncertainty* and *error* are used throughout. It would help to define what these terms actually imply; right now I read a somewhat overlapping interpretations of the terms. What is considered an error, is that a delta? E.G. a mismatch between a reference and a counterpart (or sample estimate? Or from a SRE? Or sensor observation?), and what is considered uncertainty. In case (different) errors counter act that may reduce the overall estimation error (i.e. overall mismatch), so is uncertainty of the estimate reduced then, or increased? I, myself, associate uncertainty with a random character of an error. A suggested, I think there is scope to improve respective text sentences. Findings of the study are relevant and interesting.

I read that the authors use *precipitation* but it is not very clear If precipitation other than rainfall (in liquid phase) is represented (e.g. Line 161 suggests that only rain is targeted). The title suggests precipitation, the opening sentence in the Introduction is on floods and suggests only rainfall, but it should be made more clear what precipitation actually implies as used throughout the manuscript. It is relevant to know if findings also apply to snow and hail, precipitation by frost and rhyme can be discarded.

Abstract

The (long) abstract reads a bit complicated by the large amount of information provided.

Ln 23: I struggled a bit with the conclusion. Uncertainty now becomes an error? should it not be other way around; effects of accumulated/combined errors result in ((un)predictable) uncertainty.

Introduction:

Ln. 26: Opening statement on "uniform" distributions is questionable as, especially, for extreme events we often seek information at highest time and space intervals to best represent real world rainfall forcing. Does the statement not reflect more on use of rainfall data sources, that are obtained at fixed discrete time and space steps/units (I read here "uniformly").

Lns 32 and 38: Please limit to only 2 (or max 3) most relevant references (unless truly meaningful) (please revisit the manuscript)

Lns 47-54: The GPROF algorithm is introduced and briefly described. It would be meaningful to comment on weaknesses of the algorithm in the discussion of results. The way the algorithm operates has quite some limitations that to some extend effects findings of the study.

Ln 106: Section 2.1.2 . It is not clear why in-situ data was not used directly so to perform a direct *point to (SRE) pixel* comparison. The radar estimates also is a precipitation (or rainfall) product that has errors as described but that effect on the research findings remains now undefined. Actually, a lot of findings revert back to aspects of performance of the KNMKI radar (e.g. line 285), although the radar serves as a

refence to replace in-situ gauge observations. Throughout the manuscript there are descriptions that address performance aspects of the ground radar, and at the same time GPROF, and thus aspects of performance of both sensor products are intercompared. I think this aspect of the methodological approach at the base of the study can be improved. It only takes few sentences.

Ln 106: I indicate line 106 but the comment applies to the manuscript. What also could be further clarified is on how many spatial units the analysis are based. The KNMI ground radar operates at 1km$^2$ resolution. The footprints are of different number, size and shape. I found it difficult to read how the comparisons are actually done, e.g. it is not very clear what the sample sizes are at the back of findings in figures. Often descriptions halt by simply stating, *sampling*.

Ln 115: Simply add few words what precipitation height actually implies. Please clarify on these types of terms throughout the paper to increase readability.

Ln 155: Eq. 2 is similar to Eq 1 except errors are taken as absolute. So Eq. 2 also is a relative error but now as absolute RE. A mean value (division by N is required) is not performed, NMAE simply is/becomes ARE (absolute relative error). Also check the units for consistency.

Ln 171: Why to mention explicitly "independent", is that not trivial?

Ln 182:  To relate foot print size to number of observations (this reads like sampling frequency) to threshold is quite speculative I would say. To write it differently "Is there any prove that threshold has relation to the foot print size to affect the number of observations" (Do you mean larger size of the foot print results in less spatial units for analysis)? To work with mean values is not a problem.

Ln 196: Here the "ash-tray" term *sampling* is used *bit I guess what sampling* actually complies in this study needs a better description. We have scale of sampling (the foot print), the frequency of sampling, the way a sample is taken (the sampling method), and the sample size (nr. Of observations). I advise to go through the paper again to read and to further clarify on these issues so to create better understanding. In my opinion there is scope to improve readability. Also, the term observation(s) is frequently used (e.g. annotation Fig 5 ) but does an observation now also have the meaning of a single sample?

Fig 3: It is not clear how "volume" is defined? Make clear what the meaning of volume is. Also in the writing of the paragraph (Section 3.1).

Fig 3: The legend with respective CDFs is undefined in the text. Explain what the terms *CDF GPROF; CDF 19 GHz; CDF GPRO-own and CDF 19 GHz-own* actually represent to clarify on information by the CDFs.

Fig 4: Not clear why the number of "4" methods is mentioned? The figure shows results of 3 methods applied at two heights.

Line 256: Figure 2 does not show an (comparative) numbers (or reference) to the statement. The word "likely" is used three times re argue for findings but the term actually has a dimension of "speculation". I'm not in favor of such wording as research aims to create clarify.

Ln 285: This reads difficult. How can ETH influence performance of GPROF? Ground radar is used as a reference.

Kind regards,

Tom Rientjes

---

## Author Response (AR1)

*Response to Anonymous Referee #1*
*The comments are addressed (in red) below. The line numbers correspond to the initial version of the manuscript.*

This manuscript studies the uncertainty of precipitation estimates over the Netherlands computed from radiometers' observations regarding different samplings and retrieval methods.

The precipitation estimates from radiometers are evaluated against ground-based weather radar precipitation estimates using different satellite footprints and sensors.

Finally, the methods and results are clearly described and discussed throughout the manuscript, providing new insights into the performance of the Goddard Profiling algorithm (GPROF).

The manuscript fits the scope of the AMT journal; it is well written -clear and concise- and I consider it a great contribution to the remote sensing community. Congrats!

*We highly appreciate these kind words and would like to thank the reviewer for the time to provide feedback on our manuscript.*

I have a few specific questions/comments to improve clarity.

1. Lines 29-30. What do the authors mean by "Their spatial coverage and representation, however, is limited"? Although the spatial coverage of space-borne sensors is greater than ground-based weather radars and rain gauges, this coverage is still somewhat limited (~885 km), right?

*Indeed, the coverage of radiometers is still limited. However, this sentence refers to all spaceborne sensors, including geostationary satellites. Additionally, although the coverage might be limited, the radiometers have a global representation in contrast to the local representation of ground-based radars.*

2. Lines 35-42. These paragraphs are confusing. The authors state in Line 36 that one kind of input is the indirect observation of cloud properties. However, Lines 39-42 read that direct observations "are preferred for quantitative applications in meteorology and hydrology as precipitation retrieval from visible and infrared channels is based on cloud to precipitation relations". Maybe the authors mean "the former" in Line 41, but rephrasing may be needed. Please elaborate.

*Direct observations are preferred as these observations provide information regarding precipitation. The sensors capable of measuring these direct characteristics, however, are typically found on low Earth orbit (LEO) satellites due to their frequency channels. In contrast, channels operating within the visible (VIS) and infrared (IR) frequency ranges can be accommodated on geostationary satellites, but these frequencies only provide information about cloud properties. We revised the aforementioned lines to enhance clarity.*

3. Line 64. Consider adding "detection and accuracy OF RAIN RATES" or similar.

*We appreciate the suggestion and we will rewrite line 64 as follows: "detection of precipitation and accuracy of precipitation intensity"*

4. Line 117. Why this particular threshold? Please discuss setting this threshold to help understand the radar Echo Top Height (ETH) influence in the results.

*The threshold is chosen to ensure light precipitation rates are also detected, a precipitation type that frequently occurs in the Netherlands. A drawback of this low threshold is discussed in lines 119-122.*

5. Line 207. Please replace "The first three figures" with "Figures 1-3"

*We appreciate the suggestion and we rewrote line 207 as follows: "GPROF's performance and some first results concerning the influence of sampling are shown in Figs. 1-3"*

6. Figure 2, p17. These plots are great and depict much information; however, the markers need some improvement. For instance, it is almost impossible to notice the stars in the plot showing the number of observations. I advise using other marker fill styles, such as none for circles, changing the edge colour, or other alternatives. Also, add ETH, e.g., low ETH, med ETH, etc., to the colour scale.

*The suggestion is appreciated and was implemented to improve the clarity of Fig. 2.*

*Response to T.H.M. Rientjes*

*The comments are addressed (in red) below. The line numbers correspond to the initial version of the manuscript.*

This interesting study is well developed and executed. The topic of estimating/comparing precipitation/rainfall by different remote sensing-based data sources is quite challenging. My comments mostly are textual where further detailing and description of used terms is advised to increase readability. As an example, sampling is widely used but it is not always clear how to read the meaning of the term. E.g. sampling method, sampling pattern, sampling sensitivity, sampling area are used but remain somewhat undefined. Also the term pixels is used but also undefined in terms of size shape. It would be good to add a figure that shows how spatial units of sampling overlays the pixel structure of the SRE products. Also uncertainty and error are used throughout. It would help to define what these terms actually imply; right now I read a somewhat overlapping interpretations of the terms. What is considered an error, is that a delta? E.G. a mismatch between a reference and a counterpart (or sample estimate? Or from a SRE? Or sensor observation?), and what is considered uncertainty. In case (different) errors counter act that may reduce the overall estimation error (i.e. overall mismatch), so is uncertainty of the estimate reduced then, or increased? I, myself, associate uncertainty with a random character of an error. A suggested, I think there is scope to improve respective text sentences. Findings of the study are relevant and interesting.

*We highly appreciate dr. Tom Rientjes took the effort and time to write a constructive review. We think his feedback contains great suggestions to enhance readability. We will address his suggestions point-wise below.*

I read that the authors use precipitation but it is not very clear If precipitation other than rainfall (in liquid phase) is represented (e.g. Line 161 suggests that only rain is targeted). The title suggests precipitation, the opening sentence in the Introduction is on floods and suggests only rainfall, but it should be made more clear what precipitation actually implies as used throughout the manuscript. It is relevant to know if findings also apply to snow and hail, precipitation by frost and rhyme can be discarded.

*Thank you for bringing these inconsistencies to our attention. We aimed to use 'precipitation' as we included all forms of precipitation (e.g. rainfall, snow, hail). In our revised manuscript, we consistently refer to precipitation.*

Abstract

The (long) abstract reads a bit complicated by the large amount of information provided.

Ln 23: I struggled a bit with the conclusion. Uncertainty now becomes an error? should it not be other way around; effects of accumulated/combined errors result in ((un)predictable) uncertainty.

*We acknowledge the abstract contained a high information density and was too long. Hence, we revised and shortened the abstract. In the revised version we also included your feedback related to Line 23:*

*"The Goddard Profiling algorithm (GPROF) converts radiometer observations from Global Precipitation Measurement (GPM) constellation satellites into precipitation estimates. Typically, high-quality ground-based estimates serve as reference to evaluate GPROF's performance. To*

*provide a fair comparison, the ground-based estimates are often spatially aligned to GPROF. However, GPROF combines observations from various sensors and channels, each associated with a distinct footprint. Consequently, uncertainties related to the representativeness of the sampled areas are introduced in addition to the uncertainty when converting brightness temperatures into precipitation intensities. The exact contribution of resampling precipitation estimates, required to spatially and temporally align different resolutions when combining or comparing precipitation observations, to the overall uncertainty remains unknown. Here, we analyze the current performance of GPROF over the Netherlands during a four year period (2017-2020) while investigating the uncertainty related to sampling. The latter is done by simulating the reference precipitation as satellite footprints that vary in size, geometry, and applied weighting technique. Only GPROF estimates based on observations from the conical-scanning radiometers of the GPM constellation are used. The reference estimates are gauge-adjusted radar precipitation estimates from two ground-based weather radars from the Royal Netherlands Meteorological Institute (KNMI). Echo top heights (ETH) retrieved from the same radars are used to classify the precipitation as shallow, medium, or deep. Spatial averaging methods (Gaussian weighting vs. arithmetic mean) minimally affect the magnitude of the precipitation estimates. Footprint size has a higher impact but cannot explain all discrepancies between the ground- and satellite-based estimates. Additionally, the discrepancies between GPROF and the reference are largest for low ETH, while the relative bias between the different footprint sizes and implemented weighting methods increase with increasing ETH. Lastly, our results do not show a clear difference between coastal and land simulations. We conclude that the uncertainty introduced by merging different channels and sensors cannot fully explain the discrepancies between satellite- and ground-based precipitation estimates. Hence, uncertainties related to the retrieval algorithm and environmental conditions are found to be more prominent than resampling uncertainties, in particular for shallow and light precipitation."*

Introduction:

Ln. 26: Opening statement on "uniform" distributions is questionable as, especially, for extreme events we often seek information at highest time and space intervals to best represent real world rainfall forcing. Does the statement not reflect more on use of rainfall data sources, that are obtained at fixed discrete time and space steps/units (I read here "uniformly").

*We appreciate that the reviewer points out this unclarity. With this sentence, we tried to express the importance of global representation of precipitation estimates. Ground-based estimates especially cover the European, North-American, and parts of the Australian continents. However, as precipitation characteristics vary greatly, these observations are not representative for the ungauged areas. Spaceborne estimates, however, also provide estimates over these areas.*

*We rewrote line 26 as follows:*

*"Accurate global precipitation estimates are vital for both hydrological research and operational applications like weather forecasts and flood early water systems."*

Lns 32 and 38: Please limit to only 2 (or max 3) most relevant references (unless truly meaningful) (please revisit the manuscript)

*The number of references is limited in the revised version.*

Lns 47-54: The GPROF algorithm is introduced and briefly described. It would be meaningful to comment on weaknesses of the algorithm in the discussion of results. The way the algorithm operates has quite some limitations that to some extend effects findings of the study.

*GPROF in the context of our research is discussed in lines 253-264, 267-274, 281-284, and 286-289. Furthermore, the main aim of our research is to study the effect of sampling on rainfall estimates. Hence, we think that more focus on GPROF is out of scope for this manuscript.*

Ln 106: Section 2.1.2 . It is not clear why in-situ data was not used directly so to perform a direct point to (SRE) pixel comparison. The radar estimates also is a precipitation (or rainfall) product that has errors as described but that effect on the research findings remains now undefined. Actually, a lot of findings revert back to aspects of performance of the KNMKI radar (e.g. line 285), although the radar serves as a refence to replace in-situ gauge observations. Throughout the manuscript there are descriptions that address performance aspects of the ground radar, and at the same time GPROF, and thus aspects of performance of both sensor products are intercompared. I think this aspect of the methodological approach at the base of the study can be improved. It only takes few sentences.

*The reviewer correctly points out the inaccuracies of the radar product, which we indeed mentioned in our manuscript as we think it is important to discuss the limitations of our research. However, rain gauges are point measurements with limited spatial coverage due to the high spatiotemporal variability of precipitation. As a consequence, they inevitably introduce representation issues for a given (convective) precipitation event. Furthermore, our primary objective was to investigate the uncertainties associated with sampling precipitation estimates on various footprint sizes. This analysis could not be conducted with rain gauges due to their limited availability and representation.*

*Additionally, the radar product we implemented is gauge-adjusted. Therefore, we used the reference data with the highest available quality.*

Ln 106: I indicate line 106 but the comment applies to the manuscript. What also could be further clarified is on how many spatial units the analysis are based. The KNMI ground radar operates at 1km2 resolution. The footprints are of different number, size and shape. I found it difficult to read how the comparisons are actually done, e.g. it is not very clear what the sample sizes are at the back of indings in figures. Often descriptions halt by simply stating, sampling.

*The KNMI data, both precipitation and ETH, are "resampled" onto the satellite footprints. With 'resampling' we refer to the following procedure: we take the center coordinates of a particular satellite footprint, and use the dimensions specified in Tab. 1 to select all KNMI pixels within the specified circumference from the footprint center. When selecting the pixels, we take the orientation of the spatial ellipse into account. As a next step, we either simply "average" all selected pixels (referred to as spatial ellipse in the manuscript), take the average using Gaussian weights, or select all pixels within a circular circumference instead of an elliptical form. The threshold to distinguish between dry and wet footprints and the proximity of a footprint to the borders of the KNMI dataset affect the number of observations. Hence, this number can vary between the different sampling methods.*

*This procedure is described in lines 126-136. However, we added the following information to line 136 to enhance readability and clarity:*

*"The uncertainty associated with the procedure to align either high-resolution reference observations with one radiometer resolution or combine various sizes of radiometer footprints to retrieve one estimate is referred to as 'resampling uncertainty' in the remainder of this manuscript."*

Ln 115: Simply add few words what precipitation height actually implies. Please clarify on these types of terms throughout the paper to increase readability.

*We agree this choice of words is unclear and rewrote lines 114-116 as:*

*"Ground-based radar echo top height (ETH) data was used to classify precipitation based on its vertical extent. This classification enables to study the influence of precipitation formation height on GPROF's performance and the relationship between ETH and precipitation variability within a specific footprint size."*

Ln 155: Eq. 2 is similar to Eq 1 except errors are taken as absolute. So Eq. 2 also is a relative error but now as absolute RE. A mean value (division by N is required) is not performed, NMAE simply is/becomes ARE (absolute relative error). Also check the units for consistency.

*Both equations result in a dimensionless outcome. Additionally, division by N is not required (division of summations in both eq. 1 and 2 by N would remove 1/N in both the numerator and denominator).*

Ln 171: Why to mention explicitly "independent", is that not trivial?

*This result is not necessarily trivial. The GPROF algorithm is tuned on one of the sensors (GMI), and the other radiometers have different footprint sizes and frequency channels. Although GPROF takes these differences into account, the performance of the algorithm could still be affected by the specific sensor.*

Ln 182: To relate foot print size to number of observations (this reads like sampling frequency) to threshold is quite speculative I would say. To write it differently "Is there any prove that threshold has relation to the foot print size to affect the number of observations" (Do you mean larger size of the foot print results in less spatial units for analysis)? To work with mean values is not a problem.

*With this sentence, we wanted to highlight that the number of observations may vary between sampling methods. This variation can occur as we assessed for each footprint whether the mean value exceeded the defined threshold of 0.1 mm/hr. Consequently, the mean KNMI value might exceed this threshold for one sampling method, while in another sampling method (whether larger or smaller), it may not.*

Ln 196: Here the "ash-tray" term sampling is used bit I guess what sampling actually complies in this study needs a better description. We have scale of sampling (the foot print), the frequency of sampling, the way a sample is taken (the sampling method), and the sample size (nr. Of observations). I advise to go through the paper again to read and to further clarify on these issues so to create better understanding. In my opinion there is scope to improve readability. Also, the term observation(s) is frequently used (e.g. annotation Fig 5 ) but does an observation now also have the meaning of a single sample?

*We included these suggestions when revising our manuscript. Additionally, we included descriptions for the word uncertainty or at least make sure to explain what it refers to (as mentioned by Tom in the beginning of his review). And indeed, observation refers to a single sample.*

Fig 3: It is not clear how "volume" is defined? Make clear what the meaning of volume is. Also in the writing of the paragraph (Section 3.1).

*The following information was be added to line 200 to provide a clearer description of 'volume':*

*"The CDF illustrates the probability of observing values up to and including a particular precipitation intensity level. Although lower intensities occur more frequently in the Netherlands (solid lines), their contribution to the total amount of precipitation might be limited. Therefore, also the CDFv is shown, which considers the relative contribution of a certain precipitation intensity bin to the total amount of precipitation."*

Fig 3: The legend with respective CDFs is undefined in the text. Explain what the terms CDF GPROF; CDF 19 GHz; CDF GPRO-own and CDF 19 GHz-own actually represent to clarify on information by the CDFs.

*We acknowledge we provided limited context about the various CDFs in the manuscript. To address this, we added this information to the caption of Fig. 3:*

*"Additionally, similar to Fig. 2, the results are also shown using the footprint associated with the 19 GHz channel of SSMIS and AMSR-2 (GPROF: grey, reference: darkblue), referred to as "own" as the native footprint size of the specific sensor was used."*

*and we included the colors and line types in lines 200-204.*

Fig 4: Not clear why the number of "4" methods is mentioned? The figure shows results of 3 methods applied at two heights.

*The y-axis also represents a sampling method, namely the Gaussian circle. Hence, we refer to four methods.*

Line 256: Figure 2 does not show an (comparative) numbers (or reference) to the statement. The word "likely" is used three times re argue for findings but the term actually has a dimension of speculation". I'm not in favor of such wording as research aims to create clarify.

*We will remove the word likely, except for line 218 as we cannot proof this statement within this study. We included some numbers in lines 257-258:*

*"Yet, the POD associated with shallow events remains low (varying between 0.48-0.60) in V07."*

Ln 285: This reads difficult. How can ETH influence performance of GPROF? Ground radar is used as a reference.

*Thank you for pointing out this unclear sentence. We rephrased the line as follows:*

*"Figure 6 illustrates that the sampling method has a limited effect on both the mean and standard deviation. Instead, they appear to be correlated with ETH, as expected, since higher ETH is often associated with more convection and higher precipitation rates. Additionally, ETH seems to be a better predictor of uncertainty in GPROF precipitation estimates, while the uncertainty related to sampling is minimal, as depicted in Figs. 2 and 3."*

Kind regards, Tom Rientjes

---

## Author Response (AR2)

We are glad our manuscript is accepted for publication! As the editor requested, we slightly rewrote line 328 of the tracked/changed version of the manuscript. We made another minor textual adjustment to line 218 of the tracked/changed document (line 203 of the final version) for clarification.